# Has the Efficiency of China’s Healthcare System Improved after Healthcare Reform? A Network Data Envelopment Analysis and Tobit Regression Approach

**DOI:** 10.3390/ijerph16234847

**Published:** 2019-12-02

**Authors:** Guangwen Gong, Yingchun Chen, Hongxia Gao, Dai Su, Jingjing Chang

**Affiliations:** 1School of Medicine and Health Management, Tongji Medical College, Huazhong University of Science and Technology, Wuhan 430030, China; guangwengong@163.com (G.G.); gaohongxia@hust.edu.cn (H.G.); sudai@hust.edu.cn (D.S.); changjingjing@hust.edu.cn (J.C.); 2Research Center for Rural Health Services, Hubei Province Key Research Institute of Humanities and Social Sciences, Wuhan 430030, China

**Keywords:** healthcare system, efficiency, network data envelopment analysis, Tobit regression, healthcare reform

## Abstract

Background: A healthcare system refers to a typical network production system. Network data envelopment analysis (DEA) show an advantage than traditional DEA in measure the efficiency of healthcare systems. This paper utilized network data envelopment analysis to evaluate the overall and two substage efficiencies of China’s healthcare system in each of its province after the implementation of the healthcare reform. Tobit regression was performed to analyze the factors that affect the overall efficiency of healthcare systems in the provinces of China. Methods: Network DEA were obtained on MaxDEA 7.0 software, and the results of Tobit regression analysis were obtained on StataSE 15 software. The data for this study were acquired from the China health statistics yearbook (2009–2018) and official websites of databases of Chinese national bureau. Results: Tobit regression reveals that regions and government health expenditure effect the efficiency of the healthcare system in a positive way: the number of high education enrollment per 100,000 inhabitants, the number of public hospital, and social health expenditure effect the efficiency of healthcare system were negative. Conclusion: Some provincial overall efficiency has fluctuating increased, while other provincial has fluctuating decreased, and the average overall efficiency scores were fluctuations increase.

## 1. Introduction

The World Health Organization (WHO) measured the overall efficiency in 191 WHO member states in 2000, in which China ranked only 144. [1] Hu, Shen, and Zou observed inefficiencies in China’s health resource allocation and service delivery [2]. Health system inefficiencies manifested because increasing number of patients opted to go to the higher level or city hospitals, many resources were transferred to city hospitals, and drugs were overused and expensive. These conditions have aggravated the imbalance and contradiction between urban and rural areas, regions, and people, and have become a potential threat to social stability. Therefore, the healthcare system should be reformed.

The Chinese government formally launched the healthcare reform in March 2009. The reform mainly aims to establish a basic universal system, expand basic medical insurance programs, construct a national system for essential drugs, develop a primary healthcare service, provide equal access to urban and rural dwellers, and accelerate the reform of public hospitals [3,4].

The central and local governments increased their health investments and promoted various reform measures. The proportion of total health expenditure to gross domestic product (GDP) increased from 4.55 in 2008 to 6.36 in 2017. Per capita health expenditure increased from 1094.5 yuan in 2008 to 3783.8 yuan in 2017. Some researches evaluated the efficiency of healthcare institutions in China after the launching of the healthcare reform by conducting data envelopment analysis (DEA). DEA, a nonparametric method, has been widely used to measure the relative efficiency of decision-making unit (DUM) using multiple inputs and outputs [5,6,7,8,9,10,11]. These studies evaluated the efficiency of healthcare institutions, but the high efficiency of healthcare institutions does not always produce an efficient health outcome due to the induced demand and overuse in healthcare system.

These studies used traditional DEA models to evaluate the efficiency of different healthcare institutions. Traditional DEA models are based on the concept that production technology is a black box that transforms inputs into outputs. Many production technologies have different network structures that can be divided into several components. Some components produce outputs by using the intermediate outputs obtained from their previous components. Traditional DEA models cannot provide insights into the interrelationships of components’ inefficiencies and specific guidance to DMU managers to help them improve the DMU’s efficiency [12]. The healthcare system is a typical network production system. It can be divided into three components, namely, health input, health output, and health outcome. Human health resources, including goods and financial ones, are invested into the healthcare system to healthcare services (i.e., the first stage in the healthcare system). People recuperate or maintain their health by receiving healthcare services (i.e., the second stage in the healthcare system). Thus, the efficiency of healthcare reform in reducing the inefficiency of the healthcare system and improving health outcomes remains unclear.

Färe & Grosskopf presented the “black box” of DMU, decomposed the production system into subprocesses, and conducted network DEA to investigate the divisional and overall efficiencies of a unified framework. [13] Network DEA enables the measurement of the relative efficiency of each stage and the overall efficiency of the healthcare system and identifies the weak areas to improve the overall efficiency of the healthcare system. This study conducted network DEA method to evaluate the overall and two substage efficiencies of the healthcare system in each province in China after the implementation of healthcare reform. Moreover, Tobit regression was performed to analyze the factors that influence the overall efficiency of healthcare system in provinces.

## 2. Materials and Methods

### 2.1. Network DEA Methodology

Overall efficiency refers to the product of the efficiencies of two stages based on the two-stage DEA multiplier model proposed in [14]. Suppose inputs *X_ik_*, *i* = 1, …, *m* to produce intermediate products *Z_pk_*, *p* = 1, …, *q* and outputs Y_rk_, *r* = 1, …, *s*. The *Z_pk_,* are the outputs of stage 1 as well as the inputs of stage 2. A radial input-oriented CCR network DEA model can be formulated as
(1)$$\begin{array}{l} E_{k}=\overset{s}{\underset{r-1}{\sum }}U_{r}Y_{rk}/\overset{m}{\underset{i-1}{\sum }}V_{i}X_{ik}\\ s.t.\;\;\;\;\;\;\;\;\;\;\;\;\overset{s}{\underset{r-1}{\sum }}U_{r}Y_{rj}/\overset{m}{\underset{i-1}{\sum }}V_{i}X_{ij}\leq 1,j=1,\ldots ,n\\ \;\;\;\;\;\;\;\;\;\;\;\;\;\;\;\;\;\overset{q}{\underset{p-1}{\sum }}W_{p}Z_{pj}/\overset{m}{\underset{i-1}{\sum }}V_{i}X_{ij}\leq 1,j=1,\ldots ,n\\ \overset{s}{\underset{r-1}{\sum }}U_{r}Y_{rj}/\overset{q}{\underset{p-1}{\sum }}W_{p}Z_{pj}\leq 1,j=1,\ldots ,n\\ U_{r},V_{i},W_{p}\geq \varepsilon ,\;r=1,\ldots ,s;\;i=1,\ldots ,m;\;p=1,\ldots ,q. \end{array}$$
The set of multipliers that produces the largest $E_{k}^{1}$ while maintaining the overall efficiency score at E_k_ can be calculated using the aforementioned model.
(2)$$\begin{array}{l} E_{k}^{1}=max\overset{q}{\underset{p-1}{\sum }}W_{p}Z_{pk}\\ s.t.\;\;\;\;\;\;\;\overset{m}{\underset{i-1}{\sum }}V_{i}X_{ik}=1,\\ \overset{s}{\underset{r-1}{\sum }}U_{r}Y_{rk}-E_{k}\overset{m}{\underset{i-1}{\sum }}V_{i}X_{ik}\leq 0,\\ \overset{s}{\underset{r-1}{\sum }}U_{r}Y_{rj}-\overset{m}{\underset{i-1}{\sum }}V_{i}X_{ij}\leq 0,j=1,\ldots ,n,\\ \overset{q}{\underset{p-1}{\sum }}W_{p}Z_{pj}-\overset{m}{\underset{i-1}{\sum }}V_{i}X_{ij}\leq 0,j=1,\ldots ,n,\\ \overset{s}{\underset{r-1}{\sum }}U_{r}Y_{rj}-\overset{q}{\underset{p-1}{\sum }}W_{p}Z_{pj}\leq 0,\;j=1,\ldots ,n,\\ U_{r},V_{i},W_{p}\geq \varepsilon ,\;r=1,\ldots ,s;\;i=1,\ldots ,m;\;p=1,\ldots ,q. \end{array}$$

After calculating $E_{k}^{1}$, the efficiency of the second stage is determined by using $E_{k}^{2}=E_{k}/E_{k}^{1}$.

### 2.2. Variables and Data

Network DEA was performed using three different inputs, three intermediates and three outputs. Three different inputs were used as health care resources for the production of healthcare services. These inputs included the number of health technicians per 1000 persons, hospital beds per 1000 persons, and per capita total health expenditure. Healthcare services were used as intermediates, which included the number of outpatient department visits, number of inpatient department visits, and average in-patient days. Maternal mortality, perinatal mortality rate, and life expectancy were used as health outcome measures.

In the Tobit regression analysis, the independent variables used included environment variables, which were assumed to impact the health care performance of provinces. Table 1 presents the variables used in the models along with their definitions.

The data used in this paper was taken from the China health statistics yearbook form 2009 to 2018, and from the official websites of Chinese National Bureau databases. The variables used in this study covered the period from 2009 to 2016. The sample size was limited to 30 provinces (referred to cities and autonomous regions) because of the lack of relevant data about Tibet. Among the 30 provinces, Beijing, Tianjin, Hebei province, Liaoning province, Shanghai, Jiangsu province, Zhejiang province, Fujian province, Shandong province, Guangdong province, and Hainan province belong to the eastern region and are located in China’s relatively prosperous coastal regions. Shanxi province, Jilin province, Heilongjiang province, Anhui province, Jiangxi province, Henan province, Hubei province, and Hunan province are divided into central regions. The western region includes Inner Mongolia, Chongqing, Guangxi province, Sichuan province, Guizhou province, Yunnan province, Shaanxi province, Gansu province, Qinghai province, Ningxia, and Sinkiang.

The results of network DEA were obtained on MaxDEA 7.0 software (Beijing Realworld Software Company Ltd., Beijing, China), and the results of Tobit regression analysis were obtained on StataSE 15 software (StataCorp LLC, Texas, USA).

## 3. Results

### 3.1. Results of Network DEA

The input-oriented network DEA analysis of 30 provinces was conducted on the basis of CRS assumption. The overall efficiency and two substage efficiency scores of the healthcare system from 2009 to 2016 were obtained using two outputs, three inputs, and three intermediates.

Table 2 shows the overall efficiency scores of each province and their descriptive statistics. The number of fully efficient provinces reached three in 2009, only one from 2010 to 2012, and zero from 2013 to 2016. Although the number of fully efficient provinces of 2012 to 2016 was less than that in 2009, the average overall efficiency was higher than that in 2009.

From the overall efficiency scores of each province from 2009 to 2016, only the overall efficiency scores of Beijing increased year by year. The overall efficiency scores of Tianjin, Hebei province, Shanxi province, Liaoning province, Jilin province, Heilongjiang province, Shanghai, Zhejiang province, Fujian province, Shandong province, Guangdong province, and Ningxia showed a trend of fluctuating upward, compared with 2009. Especially the overall efficiency scores of Tianjin and Guangxi province have a remarkable growth. Although the trend is upward, but the overall efficiency scores of Jilin province, Heilongjiang province, Shanghai, Zhejiang province, Shandong province, and Ningxia were still low, below 0.8. The overall efficiency scores of Jiangxi province, Henan province, Hubei province, Hunan province, Guangxi province, Guangxi province, Chongqing, Sichuan province, Guizhou province, and Yunnan province showed a trend of fluctuating downward, compared with 2009. Although the trend was downward, the overall efficiency of Jiangxi province and Guangxi province were still high, above 0.9. The overall efficiency scores of Inner Mongolia, Jiangsu province, Anhui province, Shaanxi province, Gansu province, Qinghai province, and Sinkiang had little change and fluctuated within the range of 0.05. The overall efficiency scores of Anhui province were above 0.9, whereas that of Shaanxi province, Qinghai province, and Sinkiang were low, below 0.8.

Table 3 shows the efficiency scores of the first stage for each province and their descriptive statistics. The number of fully efficient provinces was six in 2009, decreased from 2010 to 2011, and increased to eight in 2012, which ranked first among all the years. Subsequently, the number of fully efficient provinces dropped to 5 in 2013, increased from 2014 to 2016.

The efficiency scores of first stage for Beijing, Tianjin, Hebei province, Shanxi province, Inner Mongolia, Liaoning province, Jilin province, Heilongjiang province, Shanghai, Shandong province, Hunan province, Hainan province, and Ningxia showed a trend of fluctuating upward, compared with 2009. Tianjin and Shanxi province made a remarkable growth and reached fully efficient. The efficiency scores of first stage for Jiangsu province, Hebei province, Hubei province, Chongqing, Sichuan province, Guizhou province, and Yunnan province showed a trend of fluctuating downward. Especially, the efficiency scores of Guizhou province and Yunnan province change from fully efficient drop to the values below 0.9. The efficiency scores of first stage for Anhui province, Jiangxi province, Henan province, Guangdong province, and Guangxi province fluctuated between 0.9 and 1. The efficiency scores of first stage for Jiangsu province, Zhejiang province, Fujian province, Shaanxi province, Gansu province, Qinghai province, and Sinkiang had little change, fluctuating within the range of 0.05. The efficiency scores of first stage for Sinkiang were always below 0.7.

Table 4 shows the efficiency scores of the second stage for each province and their descriptive statistics. The number of fully efficient provinces was seven from 2009 to 2010, which ranked first among all the years. Subsequently, the number of fully efficient provinces dropped post-2011.

At the second stage, Sinkiang was the only fully efficient province in the eight-year period, Fujian province, Jiangxi province, Guangxi province, Hainan province, Guangxi province, Guizhou province, and Yunnan province fluctuated between 0.9 and 1.

The efficiency scores of second stage for Beijing, Tianjin, Shanghai, Fujian province, Zhejiang province, Guangdong province, Chongqing, and Sichuan province showed a trend of fluctuating upward. Hebei province, Liaoning province, Jilin province, Heilongjiang province, Shandong province, Henan province, and Qinghai province showed a trend of fluctuating downward. The efficiency scores of second stage for Shanxi province, Inner Mongolia, Anhui province, Henan province, Hubei province, Guangdong province, Guangxi province, Shanxi province, Gansu province, and Ningxia had little change.

Figure 1 shows the visual representation of the annual average overall efficiency, average efficiency scores of the first stage, and average efficiency scores of the second stage from 2009 to 2016. The average efficiency scores of the second stage were higher than those of the first stage from 2009 to 2011. The average efficiency scores of the first stage increased after 2012, and were higher than those of the second stage between 2012 and 2015. The average efficiency scores of the second stage increased and were higher than those of the first stage in 2016.

During the eight-year period, no provinces were fully efficient in both stages, but the average overall efficiency scores wavelike increased since 2012.

### 3.2. Results of Tobit Regression Analysis

Assuming that the overall efficiency, which could accurately measure the efficiency of healthcare systems, is the product of efficiency of the first and second stages, we only analyzed the factors affecting the overall efficiency in this study. The overall efficiency scores obtained through network DEA were used as dependent variables, and Tobit regression analysis was applied to evaluate the factors that affect the efficiency of the healthcare system. Table 5 shows the estimation results of Tobit regression analysis.

The regional and government health expenditures that affect the efficiency of the healthcare system were statistically significant and positive (*p* < 0.05). Number of high education enrollments per 100,000 inhabitants, social health expenditures, number of public hospitals affect the efficiency of the healthcare system were statistically significant and negative (*p* < 0.05). Per capital GDP, personal health expenditure, and number of private hospitals affect the efficiency of the healthcare system was statistically insignificant.

## 4. Discussion

This study conducted a two-stage network DEA to assess the performance of 30 provinces from 2009 to 2016. The low efficiency scores of first stage were the main reason for low overall efficiency scores from 2009 to 2011. However, the efficiency scores of first stage were improved in 2012, which were highest and contributed the highest overall efficiency scores during the eight years. This is most likely due to China successfully achieved universal health insurance coverage in 2011, which make people with health insurance use more healthcare services [14,15,16]. The fluctuating increase of overall efficiency scores from 2012 to 2015 was mainly due to the fluctuating improvement of efficiency scores in first stage from 2012 to 2015. However, the fluctuating increase of overall efficiency scores in 2016 was mainly due to the improvement of efficiency scores of second stage. For the factors influencing the efficiency of China’s healthcare system, healthcare system efficiency and socioeconomic development were positively correlated. The efficiency of the healthcare system in the western region was lower than that in the central region, and that of the central region was lower than that of the eastern region. This condition may be because the technology and management in relatively developed areas were more advanced than that in underdeveloped areas [17]. A previous work indicated that the technical efficiency average score in the eastern region was the highest, and that in the western region was the lowest among the three regions in China.

GDP per capita has no significant impact on the efficiency of the healthcare system, and the efficiency scores of healthcare systems in provinces with a low-level education were higher than those in provinces with high-level education. These finding were inconsistent with the study of Kaya Samut & Cafrı, who found a positive relation among GDP, education, and efficiency of health system in OECD countries. [18] The findings in this study were attributed to the resource allocation and patient flow in China. The provinces with more educated population were allocated with many health resources. [19] The allocation inefficiency of health resources and patients with common and prevalent disease overcrowded large hospitals, indicating that the relationships between the healthcare system efficiency education and development degree were inverse.

We observed that government health expenditures increased the efficiency of the healthcare system, social health insurance had a negative effect on the efficiency of the healthcare system, and personal health expenditures had a negative but not significant effect on the efficiency of the healthcare system. It can be explained by the inappropriate incentives in healthcare system lead to overutilization of healthcare and thus affects the efficiency of healthcare system. Kaya Samut and Cafrı also revealed that public and private health expenditures negatively affected health care efficiency in OECD countries [20]. However, another research indicated that large public and private health care expenditures could contribute to better outcomes for improving the health care efficiency of sub-Saharan Africa. The Chinese government health expenditures include health services, basic medical insurance subsidies, and administration, population, and family planning affairs. Social health insurance expenditure refers to the enterprises’ provision of basic medical insurance for urban employees, and private health spending corresponds to the out-of-pocket health expenditure. These conditions can be explained by the inappropriate incentive mechanism in the healthcare system. The weak government financial commitment to the health sector since the mid-1980s has undermined the motivation and capacity of public hospitals to provide affordable health services, thereby causing many hospitals to run on commercial lines and physicians who act as entrepreneurs [21]. Patients who could have been treated as outpatients received services as inpatients, thereby increasing the average medical expenditure and wasting health resources [22,23]. Profit-seeking doctors would persuade their patients to acquire expensive diagnoses or treatments and drugs under the fee-for-service payment system of basic medical insurance in China [24,25]. Empirical studies in China have revealed that certain types of insurance scheme tend to increase the out-of-pocket health expenditures [26]. Inappropriate admissions mostly for children and elderly with respiratory and circulatory diseases have been performed [27,28].

A statistically significant and negative relation is found between the number of public hospitals and efficiency, and the relation between the number of private hospitals and efficiency is statistically insignificant. This finding is inconsistent with the existing studies in OECD countries. Public hospitals in OECD countries, which are not profit-oriented and do not work to improve their reputation, might negatively affect efficiency. Public hospitals in China are authorized to earn revenue and to keep and use all budgetary surpluses because the Chinese government has substantially reduced its financial commitment to public hospitals since the mid-1980s [29]. Many public hospitals and physicians encourage patients to use unnecessary healthcare services because of “information gap”. [30] Patients in China can freely choose healthcare services. Patients with common and frequently occurring diseases would visit public hospitals rather than community-level medical institutions, because most public hospitals lack qualified health professionals and heavy medical equipment. Existing studies have revealed that community–level medical institutions, which provide primary healthcare, are more effective and efficient than specialty and tertiary care hospitals. Thus, the number of public hospitals negatively affects healthcare system efficiency in China. The medical services in China are relatively insufficient, thereby resulting in low–level market competition [31], and the number of private hospitals that affect efficiency is insignificant.

This study had some important limitations. First, the evaluation of healthcare system efficiency only covered the eight-year period after the implementation of healthcare reform, and the healthcare system efficiency prior the launch of healthcare reform was not performed because of lack of relevant data. Secondly, the average life expectancy of each province is calculated at the end of each five-year plan result in the lack of data on the average life expectancy of each year. Therefore, this study uses the average life expectancy at the end of the five-year plan. Third, this study uses the perinatal mortality instead of frequently used infant mortality due to lack of relevant data. Forth, the efficiency scores of the first and second stages were 0.5 in this study, which may be inconsistent with the actual situation.

## 5. Conclusions and Policy Recommendations

This paper assessed the efficiency of Chinese healthcare system after the healthcare reform using a network DEA model, explored factors affecting the efficiency by Panel Tobit Analysis. The conclusions based on the analyses were summarized as follows; the average overall efficiency scores of 30 provinces were fluctuant increase after healthcare reform. Regional differences existed in efficiency of the healthcare system in China, and government health expenditure had a positive impact on efficiency. Due to the inappropriate incentive mechanism in the healthcare system, GDP, education and social health expenditure impacted efficiency in a negative way, and as the lack of the orderly hierarchical diagnosis, the number of public hospitals had a negative impact on efficiency. Then, puts forward the following proposals. First, health resources allocation should be increased in the primary healthcare service system. Primary healthcare is considered to be an effective health service, yet the distribution of Chinese health resources is significant inequality between hospitals and primary care institutions. [32] Second, Chinese government should take a powerful ways to promote the hierarchical diagnosis system. A hierarchical diagnosis means that different hospitals are responsible for specific disease treatments and patients go to see doctors in different clinics according to their disease [33], which address could the overcrowding problem of patients with common and prevalent diseases in large hospitals. Third, the Chinese Government should pay more attention to payment reform. The inappropriate payment models would lead to the waste of social health insurance and personal health expenditures which reduce the efficiency of healthcare system. Therefore, healthcare payment system should transform from a simple fee-for-service model to complex models that ensure payment with high quality and value.

## Figures and Tables

**Figure 1 ijerph-16-04847-f001:**
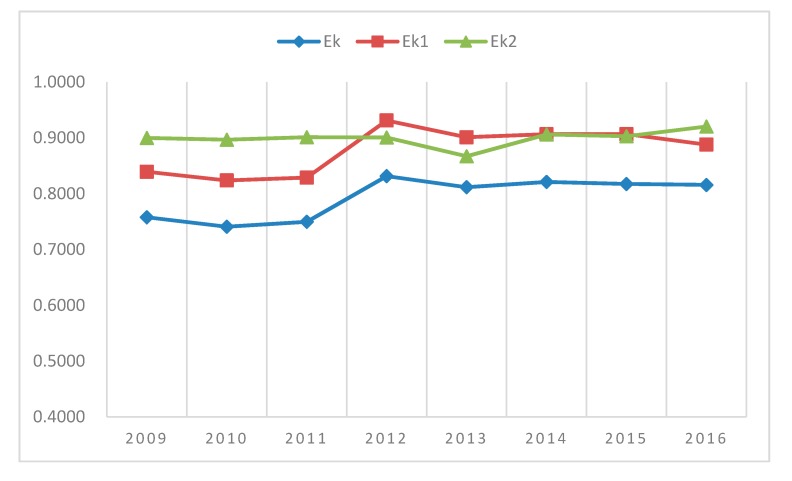
Annual averages of overall efficiency and efficiency for the sub-stages.

**Table 1 ijerph-16-04847-t001:** Model variables.

Variables	Definition	Measurement
*Inputs*		
Health technicians per 1000 Persons	Health technicians include doctors with authorization, registered nurses, pharmacists, laboratory physician, radiologists, and other medical professionals.	Density per 1000 population
Beds per 1000 Persons	Beds are the total number of beds that are available in medical institution, excluding beds for observation, extra beds, etc.	Density per 1000 population
Per capital total health expenditure	The ratio of the total amount of money raised from the whole society for health services to the population of a region in a certain period of time.	Yuan
*Outputs*		
Maternal mortality rate	The maternal mortality rate is the number of maternal deaths per 100,000 women.	Per 100,000
Perinatal survival rate	Perinatal mortality is the number of perinatal deaths, which occurs in the second trimester, during yield and within 7 days of birth.	Per 1000
Life expectancy	Life expectancy at birth measures how long, on average, a newborn can be expected to live.	Years
*Intermediates*		
Outpatients visit	Total number of outpatient visits during the Financial year.	Per 100,000
Inpatients visit	Total number of inpatient visits during the Financial year.	Per 100,000
Inpatient days	It is calculated by the total number of days discharged patients are in bed divided by the number discharged patients.	Days
*Independent variables*		
Region	China is divided into the east region, middle region and western region, according to the geographical location and the level of economic development.	
Per capital GDP	The level of per capital GDP is measured by dividing the gross domestic productof each province by the average population	Billion yuan
Education	Education level is reflected by comparing the number of higher education students per 100,000 population in each province with the average of higher education students per 100,000 population.	Per 100,0000 population
Government health expenditure	Government health expenditure are the funds used by governments at all levels for medical and health services, medical security subsidies, health and medical security administration, population and family planning affairs expenditures, and other undertakings.	10,000 yuan
Social health expenditure	Social health expenditure is outside government expenditure: all sectors of society invest in health.	Billion yuan
Personal health expenditure	Personal health expenditure are the cash payment by residents when receiving various health services.	Billion yuan
public hospital	Number of public hospitals are the total number of hospitals that are owned or controlled by a government unit or other public corporations.	
private hospital	Number of private hospitals are total number of hospitals that are not owned or controlled by government or other public organizations.	

**Table 2 ijerph-16-04847-t002:** The overall efficiency scores of 30 provinces and their distributions.

Provinces	2009	2010	2011	2012	2013	2014	2015	2016
Beijing	0.3500	0.3660	0.3982	0.6715	0.6988	0.7121	0.7186	0.7234
Tianjin	0.5623	0.5458	0.5971	0.8854	0.8739	0.9016	0.8840	0.9415
Hebei province	0.8305	0.7593	0.7922	0.8798	0.8582	0.8927	0.8772	0.8794
Shanxi province	0.6959	0.7516	0.7675	0.7747	0.7768	0.7678	0.8078	0.8315
Inner Mongolia	0.7353	0.6864	0.6750	0.7585	0.7182	0.7299	0.7306	0.7498
Liaoning province	0.5713	0.5606	0.6214	0.7059	0.6804	0.7375	0.7928	0.7467
Jilin province	0.6299	0.6259	0.6681	0.7836	0.7678	0.7912	0.7847	0.7802
Heilongjiang province	0.6631	0.6232	0.6542	0.7679	0.7388	0.7762	0.7945	0.7940
Shanghai	0.4201	0.3619	0.3891	0.7368	0.7247	0.7517	0.7249	0.7575
Jiangsu province	0.7197	0.7121	0.7033	0.7959	0.7546	0.7561	0.7476	0.7459
Zhejiang province	0.6891	0.6728	0.6823	0.7877	0.7915	0.7897	0.7634	0.7709
Anhui province	0.9454	0.9258	0.9289	0.9271	0.9375	0.9378	0.9455	0.9642
Fujian province	0.8401	0.7970	0.7831	0.8901	0.8263	0.8624	0.8648	0.9114
Jiangxi province	1	0.9493	0.9514	0.9525	0.9459	0.9362	0.9478	0.9457
Shandong province	0.7065	0.6909	0.6946	0.7233	0.7225	0.7841	0.7749	0.7879
Henan province	0.8743	0.8332	0.8586	0.8369	0.8537	0.8640	0.8457	0.8466
Hubei province	0.8131	0.7961	0.7562	0.7827	0.7398	0.7662	0.8487	0.7051
Hunan province	0.8829	0.9296	0.8903	0.8723	0.8125	0.8551	0.7898	0.7922
Guangdong province	0.7636	0.7310	0.7306	0.9298	0.9299	0.9410	0.9402	0.9306
Guangxi province	1	0.9166	0.9884	0.9676	0.9344	0.9841	0.9872	0.9301
Hainan province	0.8138	0.8933	0.8777	0.9825	0.9462	0.9582	0.9130	0.9365
Chongqing	0.8627	0.8195	0.8220	0.8089	0.8257	0.8159	0.7976	0.7782
Sichuan province	0.8274	0.7896	0.8145	0.8034	0.7922	0.8013	0.7732	0.7668
Guizhou province	1	1	1	1	0.9847	0.9268	0.9103	0.8748
Yunnan province	0.9280	0.8513	0.8818	0.9474	0.8773	0.9384	0.8840	0.8809
Shaanxi province	0.6969	0.7670	0.6825	0.7152	0.6956	0.7055	0.6997	0.7016
Gansu province	0.8811	0.8374	0.8315	0.8578	0.8355	0.8588	0.8519	0.8501
Qinghai province	0.7128	0.7160	0.7590	0.8901	0.9391	0.7075	0.7190	0.7366
Ningxia	0.6981	0.7090	0.7241	0.8402	0.7477	0.7453	0.7557	0.7600
Sinkiang	0.6194	0.6066	0.5629	0.6719	0.6192	0.6360	0.6479	0.6538
Degree of efficiency = 1	3	1	1	1	0	0	0	0
Between mean and 1	12	15	15	13	14	13	13	13
<Mean	15	14	14	16	16	17	17	17
mean	0.7578	0.7408	0.7495	0.8316	0.8116	0.8210	0.8174	0.8158
Maximum	1	1	1	1	0.9847	0.9841	0.9872	0.9642
Minimum	0.3500	0.3619	0.3891	0.6715	0.6192	0.6360	0.6479	0.6538
standard deviation	0.1596	0.1546	0.1488	0.0936	0.0959	0.0916	0.0850	0.0860

**Table 3 ijerph-16-04847-t003:** The efficiency scores of first stage for 30 provinces and their distributions.

Provinces	2009	2010	2011	2012	2013	2014	2015	2016
Beijing	0.5690	0.5972	0.5856	0.9123	0.8532	0.8064	0.8190	0.8041
Tianjin	0.6512	0.6333	0.6984	1	1	1	1	1
Hebei province	0.8305	0.7593	0.8207	0.9348	0.9243	0.9401	0.9331	0.9236
Shanxi province	0.8408	0.8710	0.9784	0.9742	0.9640	0.9538	1	1
Inner Mongolia	0.7850	0.7328	0.7843	0.8764	0.8262	0.8218	0.8438	0.8459
Liaoning province	0.7117	0.6969	0.7788	0.8918	0.8547	0.9099	0.9781	0.8977
Jilin province	0.6463	0.6383	0.7039	0.8439	0.8351	0.8561	0.8599	0.8307
Heilongjiang province	0.7654	0.7226	0.8116	0.9497	0.9018	0.9802	0.9703	0.9652
Shanghai	0.5552	0.4871	0.4872	0.8900	0.8374	0.8634	0.8405	0.8029
Jiangsu province	0.9872	0.9929	0.8563	0.9711	0.8784	0.9306	0.8540	0.8634
Zhejiang province	0.9028	0.8992	0.8667	1	0.9411	0.9071	0.8888	0.8785
Anhui province	0.9500	0.9303	0.9711	1	1	1	1	0.9995
Fujian province	0.8759	0.8304	0.7831	0.8901	0.8263	0.8624	0.8648	0.9114
Jiangxi province	1	0.9493	0.9707	1	1	0.9796	1	1
Shandong province	0.7264	0.7447	0.7239	0.79457	0.8119	0.8636	0.8419	0.8292
Henan province	1	0.9951	1	1	1	1	1	1
Hubei province	0.9150	0.9211	0.8706	0.9150	0.8565	0.8603	1	0.8083
Hunan province	0.9360	0.9837	0.9650	0.9346	0.8889	0.9293	0.8639	1
Guangdong province	1	0.9333	0.7658	0.9993	0.9861	1	1	1
Guangxi province	1	0.9166	1	0.9899	0.9523	1	0.9893	0.9301
Hainan province	0.8138	0.8933	0.9076	1	1	1	0.9504	0.9365
Chongqing	0.9588	0.9108	0.9255	0.9552	0.9311	0.8825	0.8499	0.8194
Sichuan province	0.9780	0.9367	0.9085	0.9244	0.9243	0.9155	0.9219	0.9157
Guizhou province	1	1	1	1	0.9847	0.9268	0.9103	0.8819
Yunnan province	1	0.9291	0.8921	1	0.9356	1	0.9267	0.8809
Shaanxi province	0.7775	0.8582	0.7913	0.8367	0.7883	0.7839	0.7600	0.7573
Gansu province	0.9288	0.8701	0.8960	0.9547	0.9327	0.9639	0.9675	0.9120
Qinghai province	0.7128	0.7160	0.7590	0.9199	0.9391	0.7907	0.8004	0.7921
Ningxia	0.7406	0.7538	0.7986	0.9067	0.8420	0.8339	0.9190	0.7991
Sinkiang	0.6194	0.6066	0.5629	0.6719	0.6192	0.6360	0.6479	0.6538
Degree of efficiency = 1	6	1	3	8	5	7	7	6
Between mean and 1	10	17	12	9	11	11	13	9
<Mean	14	12	15	13	14	12	10	15
mean	0.8393	0.8237	0.8288	0.9312	0.9011	0.9066	0.9067	0.8880
Maximum	1	1	1	1	1	1	1	1
Minimum	0.5552	0.4871	0.4872	0.6719	0.6192	0.6360	0.6479	0.6538
standard deviation	0.1431	0.1394	0.1329	0.0749	0.0846	0.0861	0.0863	0.0880

**Table 4 ijerph-16-04847-t004:** The efficiency scores of second stage for 30 provinces and their distributions.

Provinces	2009	2010	2011	2012	2013	2014	2015	2016
Beijing	0.6151	0.6129	0.6800	0.8190	0.7385	0.8830	0.8774	0.8997
Tianjin	0.8634	0.8619	0.8549	0.8739	0.7881	0.9016	0.8840	0.9415
Hebei province	1	1	0.9653	0.9285	0.8457	0.9495	0.9401	0.9521
Shanxi province	0.8276	0.8629	0.7845	0.8058	0.7802	0.8051	0.8078	0.8315
Inner Mongolia	0.9367	0.9367	0.8606	0.8694	0.8440	0.8882	0.8658	0.8864
Liaoning province	0.8028	0.8045	0.7978	0.7961	0.7398	0.8105	0.8105	0.8318
Jilin province	0.9746	0.9805	0.9492	0.9193	0.8487	0.9241	0.9126	0.9392
Heilongjiang province	0.8664	0.8625	0.8060	0.8192	0.7771	0.7919	0.8188	0.8227
Shanghai	0.7568	0.7429	0.7986	0.8654	0.7837	0.8706	0.8625	0.9435
Jiangsu province	0.7290	0.7172	0.8213	0.8591	0.8315	0.8124	0.8753	0.8639
Zhejiang province	0.7633	0.7482	0.7872	0.8410	0.8045	0.8705	0.8589	0.8775
Anhui province	0.9951	0.9951	0.9565	0.9375	0.9241	0.9378	0.9455	0.9647
Fujian province	0.9592	0.9598	1	1	0.9720	1	1	1
Jiangxi province	1	1	0.9802	0.9459	0.9285	0.9557	0.9478	0.9457
Shandong province	0.9727	0.9278	0.9595	0.8898	0.8534	0.9080	0.9204	0.9502
Henan province	0.8743	0.8374	0.8586	0.8537	0.8509	0.8640	0.8457	0.8466
Hubei province	0.8886	0.8642	0.8686	0.8637	0.8501	0.8906	0.8487	0.8723
Hunan province	0.9434	0.9450	0.9226	0.9141	0.8943	0.9201	0.9142	0.7922
Guangdong province	0.7636	0.7832	0.9540	0.9430	0.9148	0.9410	0.9402	0.9306
Guangxi province	1	1	0.9884	0.9812	0.9198	0.9841	0.9979	1
Hainan province	1	1	0.9670	0.9462	0.9045	0.9582	0.9607	1
Chongqing	0.8997	0.8997	0.8882	0.8868	0.8589	0.9245	0.9385	0.9497
Sichuan province	0.8460	0.8430	0.8966	0.8570	0.8252	0.8752	0.8387	0.8374
Guizhou province	1	1	1	1	1	1	1	0.9919
Yunnan province	0.9280	0.9163	0.9884	0.9376	0.9384	0.9384	0.9538	1
Shaanxi province	0.8964	0.8938	0.8625	0.8825	0.8699	0.9000	0.9206	0.9265
Gansu province	0.9487	0.9624	0.9280	0.8958	0.8763	0.8909	0.8805	0.9321
Qinghai province	1	1	1	1	1	0.8948	0.8984	0.9298
Ningxia	0.9426	0.9406	0.9067	0.8880	0.8476	0.8938	0.8223	0.9511
Sinkiang	1	1	1	1	1	1	1	1
Degree of efficiency = 1	7	7	4	4	3	3	3	5
Between mean and 1	9	10	11	9	10	16	10	14
<Mean	14	13	15	17	17	11	17	11
mean	0.8998	0.8966	0.9010	0.9007	0.8670	0.9062	0.9029	0.9204
Maximum	1	1	1	1	1	1	1	1
Minimum	0.6151	0.6129	0.6800	0.7961	0.7385	0.7919	0.8078	0.7922
standard deviation	0.1001	0.1013	0.0835	0.0597	0.0730	0.0563	0.0589	0.0610

**Table 5 ijerph-16-04847-t005:** Random effects Tobit regression results.

Parameter Regions	Coef.	Std. Error	*p*	95% [Conf. Interval]	
Regions	0.055156	0.010163	0.000	0.035237	0.075074
Per capital GDP	−0.006501	0.005211	0.212	−0.016714	0.003711
Number of high education enrollment	−0.000070	0.000009	0.000	−0.000087	-0.000053
Governmen health expenditure	0.000821	0.000101	0.000	0.000624	0.001019
Social health expenditure	−0.000233	0.000050	0.000	−0.000330	-0.000136
Personal health expenditure	−0.000171	0.000087	0.050	−0.000341	0.000000
Number of public hospitals	−0.000212	0.000046	0.000	−0.000301	-0.000123
Number of private hospitals	−0.000063	0.000033	0.057	−0.000127	0.000002
cons	0.904079	0.029746	0.000	0.845779	0.962379

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
