# Peer review of "Has the Efficiency of China’s Healthcare System Improved after Healthcare Reform? A Network Data Envelopment Analysis and Tobit Regression Approach"

_ijerph, 2019, doi:10.3390/ijerph16234847_

Round 1

Reviewer 1 Report

strength :this is a big data for discussion of the government expenditure effect the efficiency of the health system in a positive way.

 I suggest some proposal 

1 The conclusion to  be more synopsis

2 The limitation shift to the other part . Not in the conclusion part 

Author Response

Dear Reviewer:

Thank you for your reviewer’ comments concern our manuscript entitle “Has the Efficiency of Health Care System in China Improved after Healthcare Reform? A Network Data Envelopment Analysis and Tobit Regression Approach” (Manuscript ID: ijerph-637920).Those comment are all valuable and helpful foe revising and improving our paper. We have studied comments carefully and have made correction which we hope meet with approval. Revised portion are highlighted in yellow and in the "Track Changes" function in Microsoft Word. The main corrections in the paper and the responds to the reviewer’ comments are as flowing:

1. I suggest some proposal

Response: We have added some proposal according to the Reviewer’s comments.

2. The conclusion to be more synopsis

Response: We have re-written this section according to the Reviewer’s comments.

3. The limitation shift to the other part. Not in the conclusion part

Response: We have made correction according to the Reviewer’s comments.

Special thanks to you for your good comments.

We tried our best to improve the manuscript and made some changes in the manuscript. We appreciate for your warm work earnestly, and hope that the correction will meet with approval.

Reviewer 2 Report

General: The authors have identified an interesting research question. “Has the Efficiency of Health Care System in China Improved after Healthcare Reform? A Network Data Envelopment Analysis and Tobit Regression Approach” is an interesting topic.

The title is appropriate.

Abstract:

Expand abbreviations on first use in abstract

Line 16-17 “This paper take network data envelopment analysis to…” change to “This paper utilized network data envelopment analysis to…”

Conclusion sentence is confusing, please use proper grammar.

Introduction:

Expand abbreviations on first use in manuscript

Line 46: what is renminbi? Yuan? Please mention as all users might know it.

Please explain about DEA? What is it and how it helps in the introduction section

Methods:

Any confounders to the study and how they are excluded should be mentioned

Results:

The statistical analysis are complete and well conducted

Discussion Section:

Authors mention “The efficiency of the health care system in the western region was lower than that in the central region, and that of the central region was lower than that of the eastern region. This condition may be because the technology and management in relatively developed areas were more advanced than that in underdeveloped areas”….please explain which are developed and underdeveloped areas in china.

Limitations should be at the end of discussion section and not in conclusion section.

Tables are appropriate.

Overall a well conducted study.

English and grammar need significant improvement as there are lot of grammatical errors and some sentences are hard to understand. I would recommend the authors to get this manuscript evaluated by a professional English language improvement and improve its quality.

Author Response

Dear Reviewer:

Thank you for your reviewer’ comments concern our manuscript entitle “Has the Efficiency of Health Care System in China Improved after Healthcare Reform? A Network Data Envelopment Analysis and Tobit Regression Approach” (Manuscript ID: ijerph-637920).Those comment are all valuable and helpful foe revising and improving our paper. We have studied comments carefully and have made correction which we hope meet with approval. Revised portion are highlighted in yellow and in the "Track Changes" function in Microsoft Word. The main corrections in the paper and the responds to the reviewer’ comments are as flowing:

1. The title is appropriate.

Response: Thank you for your comment.

2. Abstract: Expand abbreviations on first use in abstract

Response: We have made correction according to the Reviewer’s comments.

3. Line 16-17 “This paper take network data envelopment analysis to…” change to “This paper utilized network data envelopment analysis to…”

Response: We have made correction according to the Reviewer’s comments.

4. Conclusion sentence is confusing, please use proper grammar.

Response: We are very sorry for the conclusion is confusing, and we have re-written this section.

5. Introduction: Expand abbreviations on first use in manuscript.

Response: We have made correction according to the Reviewer’s comments.

6. Line 46: what is renminbi? Yuan? Please mention as all users might know it.

Response: We have made correction according to the Reviewer’s comments.

7. Please explain about DEA? What is it and how it helps in the introduction section

Response: We have added the explanation about DEA in the introduction section according to the Reviewer’s comments.

8. Methods: Any confounders to the study and how they are excluded should be mentioned

Response: In this study, the results of Test Between-Subject Effect pointed that the interaction of the influencing factors was not significant, so there was no confounding factor.

9. Results: The statistical analysis are complete and well conducted

Response: Thank you for your comment.

10. Discussion Section: Authors mention “The efficiency of the health care system in the western region was lower than that in the central region, and that of the central region was lower than that of the eastern region. This condition may be because the technology and management in relatively developed areas were more advanced than that in underdeveloped areas”….please explain which are developed and underdeveloped areas in china.

Response: China has 31 provincial-level administrative units (hereafter “provinces”), four of which are “municipalities” (Beijing, Tianjin, Shanghai, and Chongqing). Among the 31 provinces, Beijing, Tianjin, Hebei province, Liaoning province, Shanghai, Jiangsu province, Zhejiang province, Fujian province, Shandong province, Guangdong province and Hainan province belong to the eastern region, which located in China’s relatively prosperous coastal regions. Shanxi province, Jilin province, Heilongjiang province, Anhui province, Jiangxi province, Henan province, Hubei province and Hunan province belong to central regions, which located in China’s inland regions. The western region includes Inner Mongolia, Chongqing, Guangxi province, Sichuan province, Guizhou province, Yunnan province, Shaanxi province, Tibet, Gansu province, Qinghai province, Ningxia and Sinkiang. The sample size of this paper was limited to 30 provinces (referred to cities and autonomous regions) because of the lack of relevant data about Tibet. Due to the Open-Door Policy and geographical advantage, eastern region have fast growth, which were considered as the relative developed areas. And central region shows the lower level of development than eastern region and more developed than western china.

“The efficiency of the health care system in the western region was lower than that in the central region, and that of the central region was lower than that of the eastern region. This condition may be because the technology and management in relatively developed areas were more advanced than that in underdeveloped areas”…. These sentences aims to reveal the regional disparities for efficiency of Chinese healthcare system.

11. Limitations should be at the end of discussion section and not in conclusion section.

Response: We have made correction according to the Reviewer’s comments.

12. Tables are appropriate.

Response: Thank you for your comment.

13. Overall a well conducted study.

Response: Thank you for your comment.

14. English and grammar need significant improvement as there are lot of grammatical errors and some sentences are hard to understand. I would recommend the authors to get this manuscript evaluated by a professional English language improvement and improve its quality.

Response: We are very sorry for the grammatical errors and unclear sentences. The manuscript have be modified by professional English language service.

We tried our best to improve the manuscript and made some changes in the manuscript. We appreciate for your warm work earnestly, and hope that the correction will meet with approval.

Reviewer 3 Report

Please have paper edited by native English speaker.  Many grammar mistakes detract from paper's ability to reach wide audience.

An important item is your choice of health outcomes.  They should reflect areas in which the health care system is expected to have a clear impact.  The choice of maternal mortality and perinatal mortality are good, because these are clearly impacted by health system, but life expectancy has been shown to be only very little impacted by health care system - it is mostly the social environment.  I would suggest instead using something like vaccination rates which are under the control of the system.

For non-Chinese readers, a brief description of the provinces would help put the results in context.

Author Response

Dear Reviewer:

Thank you for your reviewer’ comments concern our manuscript entitle “Has the Efficiency of Health Care System in China Improved after Healthcare Reform? A Network Data Envelopment Analysis and Tobit Regression Approach” (Manuscript ID: ijerph-637920).Those comment are all valuable and helpful foe revising and improving our paper. We have studied comments carefully and have made correction which we hope meet with approval. Revised portion are highlighted in yellow and in the "Track Changes" function in Microsoft Word. The main corrections in the paper and the responds to the reviewer’ comments are as flowing:

Please have paper edited by native English speaker. Many grammar detract from paper's ability to reach wide audience.

Response: We are very sorry for the grammatical mistakes. The manuscript have be modified by professional English language service.

An important item is your choice of health outcomes. They should reflect areas in which the health care system is expected to have a clear impact. The choice of maternal mortality and perinatal mortality are good, because these are clearly impacted by health system, but life expectancy has been shown to be only very little impacted by health care system - it is mostly the social environment. I would suggest instead using something like vaccination rates which are under the control of the system.

Response: Thank you for your comment. Life expectancy is the key metric for assessing population health, which is one of the common indexes used by WHO in measuring the health status of a country or a region. In previous studies, life expectancy has been used as one of the commonly used indicators to evaluate the output of health care system. This study focus on the health status of population in each province of China, which not only focus at mortality at a young age, but also capture the mortality along the entire life course. Life expectancy tells us the average age of death in a population. Therefore, the life expectancy has been used in this study as one indicators to measure health outcome.

For non-Chinese readers, a brief description of the provinces would help put the results in context.

Response: We have added a brief description of the provinces in Variables and Data section according to the Reviewer’s comments.

We tried our best to improve the manuscript and made some changes in the manuscript. We appreciate for your warm work earnestly, and hope that the correction will meet with approval.

Round 2

Reviewer 2 Report

The authors have made changes as requested by the reviewers.

I still see some English and grammatical errors in the manuscript mainly in the discussion section where the authors have added new paragraphs. These will need to be rechecked and corrected.